# Multi-Stage Platform for (Semi-)Automatic Planning in Reconstructive Orthopedic Surgery

**DOI:** 10.3390/jimaging8040108

**Published:** 2022-04-12

**Authors:** Florian Kordon, Andreas Maier, Benedict Swartman, Maxim Privalov, Jan Siad El Barbari, Holger Kunze

**Affiliations:** 1Pattern Recognition Lab, Friedrich-Alexander University Erlangen-Nuremberg, 91058 Erlangen, Germany; andreas.maier@fau.de (A.M.); holger.hk.kunze@siemens-healthineers.com (H.K.); 2Erlangen Graduate School in Advanced Optical Technologies (SAOT), Friedrich-Alexander University Erlangen-Nuremberg, 91052 Erlangen, Germany; 3Advanced Therapies, Siemens Healthcare GmbH, 91031 Forchheim, Germany; 4Department for Trauma and Orthopaedic Surgery, BG Trauma Center, Ludwigshafen, 67071 Ludwigshafen, Germany; benedict.swartman@bgu-ludwigshafen.de (B.S.); maxim.privalov@bgu-ludwigshafen.de (M.P.); jan.elbarbari@bgu-ludwigshafen.de (J.S.E.B.)

**Keywords:** computer-assisted surgery, surgical planning, reconstructive orthopedic surgery, ligament reconstruction, deep learning, multi-task learning, X-ray images

## Abstract

Intricate lesions of the musculoskeletal system require reconstructive orthopedic surgery to restore the correct biomechanics. Careful pre-operative planning of the surgical steps on 2D image data is an essential tool to increase the precision and safety of these operations. However, the plan’s effectiveness in the intra-operative workflow is challenged by unpredictable patient and device positioning and complex registration protocols. Here, we develop and analyze a multi-stage algorithm that combines deep learning-based anatomical feature detection and geometric post-processing to enable accurate pre- and intra-operative surgery planning on 2D X-ray images. The algorithm allows granular control over each element of the planning geometry, enabling real-time adjustments directly in the operating room (OR). In the method evaluation of three ligament reconstruction tasks effect on the knee joint, we found high spatial precision in drilling point localization (ε<2.9mm) and low angulation errors for k-wire instrumentation ($\varepsilon <0.75^{\circ }$) on 38 diagnostic radiographs. Comparable precision was demonstrated in 15 complex intra-operative trauma cases suffering from strong implant overlap and multi-anatomy exposure. Furthermore, we found that the diverse feature detection tasks can be efficiently solved with a multi-task network topology, improving precision over the single-task case. Our platform will help overcome the limitations of current clinical practice and foster surgical plan generation and adjustment directly in the OR, ultimately motivating the development of novel 2D planning guidelines.

## 1. Introduction

Joint injuries and the following structural instability are challenging patterns in orthopedics, usually caused by events of dislocation or unnatural motion and tears of restraining ligaments [1,2,3,4]. The destabilizing event comes with various implications for the patient’s health and well-being, causing significant joint pain, functional impairment, symptomatic giving way, osteochondral lesions, or damage to the articular cartilage [3,5,6,7,8,9,10,11,12,13]. For particularly severe lesions, the biomechanically correct joint function is restored by surgical reconstruction to relieve the patient’s symptoms and restore long-term stability [1,7,12,14,15,16,17,18,19,20].

An essential instrument to elevate the safety and accuracy of surgical reconstruction is the careful planning and outcome verification on 2D image data. Fundamentally, a surgical plan involves interrelating several salient features of the patient anatomy, such as osseous landmarks and anatomical axes, and performing a geometric construction for the underlying task. For example, reconstruction surgery of the Medial Patellofemoral Ligament (MPFL) necessitates a precise fixation of the substitute graft on the femoral surface. Specifically, the physiological correct ligament insertion point that guarantees joint isometry can be approximated by the construction of the Schoettle Point [21] on a 2D true-lateral radiograph. This construction builds on two anatomical keypoints on the distal femur surface and a subsection of the posterior cortex of the femoral shaft to ultimately localize the optimal point for tunnel drilling. Another application is the careful placement and angulation of k-wires inserted into the patient anatomy for transtibial Posterior Cruciate Ligament (PCL) reconstruction surgery, guiding the subsequent tunnel drilling [22,23,24,25]. High precision in the tunnel’s placement is essential to prevent any defects in joint biomechanics after reconstruction and unnecessary graft abrasion [26]. It also protects from early breakout of the bone cortex with the risk of damaging nearby vascular structures [27,28,29,30,31].

For such reconstructive applications, surgical planning promises elevated treatment safety and standardized assessment with a physiologically-grounded rationale. However, several clinical and technical challenges complicate the plan’s intra-operative application.

Most planning methods rely on the combination of salient proxy structures. This indirect description implies a greater number of error sources, which translates to higher observer-variability if the planning is done manually;Certain surgical planning can have a high level of geometric complexity. Sufficiently precise manual execution is only possible with tailored software tools or otherwise requires great amounts of time and labor;The ability to register pre-operative planning and intra-operative live data is highly complex due to the variable configuration of the joint and arbitrary relation between patient, table, and imaging system;Ad-hoc modifications of the surgical plan are essential to compensate for motion during the intervention;Manual interaction with a computer-assisted planning system is undesirable due to the surgery’s sterile setting. At the same time, the planning system should offer granular controls to correct each construction step with real-time visualization.

We propose a versatile platform for (semi-)automatic 2D planning and verification in reconstructive orthopedic surgery that addresses most of the above challenges. Our method involves three sequential stages: (i) First, a deep learning system detects the relevant anatomical structures for the underlying surgical task on 2D image data. The detection problem for each type of anatomical feature is solved in parallel using a multi-task learning (MTL) scheme; (ii) Second, the identified structures are transferred to geometric representations, such as keypoint coordinates and line segments. If needed, post-processing can be used to derive additional information required by the planning geometry; (iii) Third, the objects are combined and interrelated using a pre-defined blueprint describing the planning geometry and calculation of the planning result. By mimicking each manual step of the clinically established procedure, we align the automatic planning process to clinical practice as close as possible. This interpretable relation between anatomy and planning geometry reduces the hurdle of acceptance that the practitioner must overcome. Additionally, if needed, the segmentation of the relevant bone can be used for a registration of the planning result on subsequent live images. This allows an easier transfer of the approach into an intra-operative workflow.

We evaluate our approach on three kinds of ligament reconstruction surgery on the knee joint (Figure 1): (i) Localization of the optimal graft fixation point (Schoettle Point) for tunnel drilling in MPFL reconstruction surgery [21]; (ii) Localization of anteromedial (AM) and posterolateral (PL) bundle attachment points on the femur and tibia for Anterior Cruciate Ligament (ACL) reconstruction [32,33,34]; (iii) Finding the optimal guidewire orientation and insertion point for transtibial PCL reconstruction [23,24]. We study different strategies for separating the anatomical features into individual tasks for each application and evaluate different levels of parameter sharing using multi-head, multi-decoder, and single-task network topologies. Furthermore, we analyze a shared inductive bias across all three applications and compare the performance to the single-application case.

## 2. Related Work and Contribution

### 2.1. Image-Based Surgical Planning in Orthopedics and Traumatology

Prior research generally puts image-based surgical planning as a central component of computer-assisted orthopedic surgery and a driver for increased safety and minimal invasiveness of surgical intervention [35,36,37]. We refer the reader to comprehensive reviews covering general concepts and applications in robotics [38,39,40,41,42,43], navigation [44,45,46], surgical training [47], and augmented reality in orthopedic surgery [48].

The overall process of image-based surgical planning can be structured into four main components: (1) (Interactive) Visualization, (2) Modeling, (3) Analysis, and (4) Plan generation [49]. Visualization addresses intuitive ways of enriching the original image with relevant numeric and geometric information for the application [48]. This information includes the rotational and positional relation of joints [50], anatomic landmarks [51,52], and optimal trajectories for instrumentation and implant positioning [53,54,55,56,57]. Modeling covers the selection and mathematical representations of these data, allowing subsequent optimization of the planning solution, i.e., numerical and geometric analysis. The academic community has explored these two steps mainly by conventional image analysis [36], statistical shape modeling and machine learning [58], and, more recently, via deep learning [59]. The prevailing representation and analysis methods are semantic segmentation of the target surface in 3D, landmarking of salient regions using numerical keypoints, the fusion of image information with simulated joint kinematics [39,60], and registration using camera data, markers, or anatomic surface information [48].

Despite the wide adoption of pre-operative plans in various types of orthopedic surgery, their application to trauma interventions, such as ligament reconstruction, is limited [35,37,48]. For these cases, diagnostic 3D-scans are rarely used and require complex 3D/2D registration of localized intra-operative fluoroscopic data with the pre-operative volume. On the other hand, planning on diagnostic 2D radiographs has limited value in the operating room due to limitations in patient positioning and the arbitrary relation between the patient, table, and imaging device. In these scenarios, intra-operative planning on 2D image data become an increasingly attractive alternative [21,61].

### 2.2. Multi-Task Learning and Task Weighting

In MTL, data samples from multiple related tasks are used together to build a learning system with shared inductive bias [62]. By leveraging similarities in data and their underlying formation process, the goal is to discover more generalized and potent features that ultimately lead to better performance than independent single-task learning (STL) inference [62]. For feature-based MTL, the shared inductive bias is typically realized by soft or hard parameter sharing [62,63,64,65]. In the soft case, the parameters for each task are separated but constrained using a distance penalty. In hard parameter-sharing, the interaction between tasks is structured by using a meta learner with shared parameters and individual learners with task-specific parameters [66].

For most MTL problems, the update policy for the meta-learner parameters should be designed to prevent any dominance of certain tasks over others. For task balancing, several weighting heuristics have been proposed, most of them being based on a weighted summation of the task-specific losses [67]. Gradient normalization (GradNorm) dynamically approximates the gradient magnitude of the shared representation w.r.t. for each task and aims to level the individual training rates [68]. Liu et al. [69] simplify this idea by directly balancing the task losses’ rate of change over training time without considering gradient magnitude. Kendall et al. [70] propose task uncertainty as a proxy for each task’s weight, whereas Sener and Koltun [71] formulate MTL as a multi-objective optimization based on Pareto-optimal updates. Multiple solutions consider analysis and manipulation of individual gradient directions. Yu et al. [72] project the task gradients onto the normal plane of conflicting tasks, thus avoiding opposing gradient directions and establishing a stable update trajectory. Wang et al. [73] update this formulation by explicitly adapting gradients for cases of positive similarity to better capture the diverse and often highly complex task interactions. A different implementation of this rationale is probabilistic masking of gradients with a conflicting sign using a drop-out-style operation [74], or weighting the tasks such that their aggregated gradient has equal projections onto individual tasks [75].

We use GradNorm task balancing [68] in this work as it shows competitive performance for more related tasks and is easy to implement at little to no computational overhead.

### 2.3. Contribution

Our method builds on previous work [50,54,55] and extends them in several ways:This work establishes a multi-stage workflow that covers all necessary steps for image-based surgical planning on 2D X-ray images. The workflow is designed to mimic the clinically-established manual planning process, enabling granular control over each anatomical feature contributing to the planning geometry;We evaluate the model for three trauma-surgical planning applications on both diagnostic as well as intra-operative X-ray images of the knee joint. The numeric results match clinical requirements and encourage further clinical evaluation;We empirically show that the detection of anatomical landmarks benefits from a MTL setting. We confirm that explicit task weighting significantly reduces the landmark localization error, and illustrate that a multi-head network topology achieves similar performance to task-specific decoders, which are computationally much more expensive.Our study demonstrates that sharing tasks across anatomically related applications does not significantly improve performance compared to the single-application variant.

### 2.4. Article Structure

In Section 3, we present the multi-stage algorithm for (semi-)automatic orthopedic surgery planning and provide details on each step. Furthermore, we describe the different MTL schemes considered in this work, their training protocols, the pre- and intra-operative datasets used, and finally the evaluation metrics for reporting the results. In Section 4, we pose several research questions that are investigated through a series of experiments. After that, we discuss the observations and findings in Section 5 and summarize our contribution in Section 6.

## 3. Materials and Methods

### 3.1. (Semi-)Automatic Workflow for 2D Surgical Planning

#### 3.1.1. Multi-Stage Planning Algorithm

In the following, we describe a multi-stage algorithm that automatically models, analyzes, and visualizes a 2D orthopedic surgery plan. The general workflow is illustrated in Figure 2.

Before any geometric processing, a planning blueprint for the target surgical application needs to be defined, describing the necessary sequential construction steps. Based on this blueprint, we can derive the relevant anatomical features and group them according to the underlying task type. Furthermore, any post-processing that builds upon these features can be defined, e.g., the calculation of intersection points, or the extraction of relevant parts of a bone outline. In this work, we focus on three main feature and spatial task types (excluding any secondary features generated by post-processing) that are employed by most surgical planning procedures:Semantically coherent regions. Segmentation of connected regions that share certain characteristics, mostly bones and tools;Anatomical keypoints. Point-like landmarks that pinpoint features of interest on the bone surface;Elongated structures. Straight and curved lines that describe edges, ridges, or that refer to indirect features, such as anatomical axes.

After task configuration, in Stage A), an MTL algorithm extracts all relevant anatomical structures in parallel, sorted by their task type. Second, in Stage B), the features are translated from their original inference encoding to geometrically interpretable objects/descriptors. For example, the heatmap encoding of a keypoint is converted to its geometric equivalent of *x* and *y* coordinates, using the spatial argmax operation or a least-squares curve fitting. Stage C) finally describes the automatic processing of the individual planning steps and its overlay on the input image. The user can interact with the planning result to correct any misalignment, and the planning is updated in real-time. In the following, we provide details on the individual stages.

#### 3.1.2. Stage A) MTL for Joint Extraction of Anatomical Features

Following the MTL paradigm, we want to find a functional mapping $f_{\theta }^{t}:X\to Y^{t}$ from the input space X to multiple task solution spaces ${\left\{Y^{t}\right\}}_{t\in [T]}$ [62,71]. θ is a set of trainable parameters and *T* is the number of tasks. The data that are required to approximate this mapping in a supervised learning manner are characterized by *M* datapoints ${\left\{x_{i},y_{i}^{1},\ldots ,y_{i}^{T}\right\}}_{i\in [M]}$, where $y_{i}^{T}$ is the individual label for task *t*.

We further consider a hard parameter-sharing strategy and split the parameters θ into a single set of shared parameters $\theta^{sh}$ and task-specific parameter sets ${\left\{\theta^{t}\right\}}_{t\in [T]}$ [66]. Specifically, we define a meta learner $g_{\theta^{sh}}:X\to Z$ and individual learners for each task $h_{\theta^{t}}^{t}:Z\to Y^{t}$ that by their composition $h_{\theta^{t}}^{t}\circ g_{\theta^{sh}}$ yield function
(1)$$f_{\theta^{sh},\theta^{t}}^{t}:X\to Y^{t}.$$

This learning scheme allows to leverage dependencies and similarities between the tasks, and to retain task-specific discriminative capacity. Specifically, for the extraction of anatomical features, the salient structures used for localizing certain landmarks may be of similar importance when trying to solve a segmentation task of the complete bone. In addition, most structures relevant for surgical planning share the same underlying anatomical processes. For the optimization of each task with gradient descent, a distinct loss function $L^{t}\left(\;\cdot \;,\;\cdot \;\right):Y^{t}\times Y^{t}\to R_{0}^{+}$ is used. Although each loss function dictates the updates for the individual learners, the parameter updates for the meta learner are derived by their weighted linear combination. The general optimization problem can be written as
(2)$$\min_{\theta^{sh};\;\theta^{1},\ldots ,\theta^{T}}\sum_{t=1}^{T}w^{t}\left[\frac{1}{M}\sum_{i=1}^{M}L^{t}\left(f_{\theta^{sh},\theta^{t}}^{t}\left(x_{i}\right),y_{i}^{t}\right)\right].$$
$w_{t}$ is a scalar weight for the loss function for the *t*th task.

Although the choice for the task’s loss function in theory only depends on the task type, it should ideally be based on a spatial representation of the anatomical structure of interest. Such representation exploits dependencies in spatial appearance and a closer alignment of the feature topology. For example, semantic segmentation and heatmap-based landmark regression rely on a strong pixel-wise correlation between input and target. In contrast, a numerical regression task where the *x* and *y* coordinates of a landmark are approximated with a fully-connected layer typically exhibits a weaker spatial coupling between the learned features and the numerical coordinates. For this reason, we use spatial representations for the tasks considered in this work (Table 1).

The derivation of the weights $w_{t}$, which ultimately steer the task’s relative priority, is a delicate yet decisive step. In general, we can distinguish two variants: (i) Constant weights, and (ii) Dynamic weights which are adapted during optimization. At first glance, constant weights are easily interpretable and can be used to incorporate prior knowledge to assign relative importance. However, since these weights act as multiplicative factors of the gradient magnitudes, which might vary significantly across subsequent backward passes, it is difficult to identify a suitable and stable weight composition [54]. In addition, different magnitudes of the individual loss functions need to be accounted for to prevent dominance of some tasks over others. These downsides can be mitigated by using dynamic weighting policies, which compensate for fluctuations in the tasks’ relative standings. They, however, come with the disadvantage of more elaborate update algorithms, or, depending on the particular policy and task composition, might lead to inferior total performance.

For our particular problem scenario, we consider the following weighting strategies.

Uniform (constant). All tasks are weighted uniformly: $w^{t}=1$.Balanced relative learning rates (dynamic). Gradient normalization by Chen et al. [68] to ensure balanced training rates of all tasks, i.e., approximately equally-sized update steps for each task: $w^{t}=\mathrm{GradNorm}\left(t\right)$.

We further consider different variants of model topology that differ in proportion between shared and task-specific parameters. Indeed, the shared inductive bias’s capacity greatly influences the types of features learned in the meta learner. As the task-specific complexity shrinks, more general features need to be extracted, leading to superior solutions for similar tasks that benefit from each other or inferior solutions for conflicting tasks. We analyze this trade-off using two model topologies that facilitate different levels of parameter sharing and compare them to a single-task earning baseline.

Single-task baseline. For comparison, a STL baseline of independent encoder-decoder structures is optimized. Here, no parameters are shared;Multi-head topology. Both the encoder and decoder parameters are shared between the tasks in this variant. The output of the decoder is fed into task-specific prediction heads, which involve significantly less dedicated parameters and, thus, require a multi-purpose feature decoding. We argue that such a constrained decoder might benefit learning for highly similar tasks;Multi-decoder topology. After feature extraction in a shared feature encoder, the latent representation is used as input for the task-specific decoders and prediction heads. In other words, an abstract representation has to be found that serves the reconstruction for different kinds of tasks.

#### 3.1.3. Stage B) Extraction of Geometric Objects and Post-Processing

In Stage A), the multi-task network infers all anatomical features via specific spatial encodings. These encodings ensure an efficient optimization of the MTL scheme but cannot be directly used for the geometrical construction defined in the planning blueprint. For that reason, the feature representations are subsequently transformed to more general descriptors using a common set of operations shared between different planning applications.

For the three main tasks (Section 3.1.2), the representation transformations are as follows. The pixel-wise segmentation on the input image is interpreted as subsets $S_{k}$ with k∈{0,…,K} that represent coherent regions of interest, e.g., bones, implants, soft tissue, or background. Due to the multi-label definition, the subsets do not have to be disjoint, which allows to identify region overlap. This is an important trait in the context of transmissive imaging, which assumes an additive model of the pixel intensities and, thus, perceivable superimposition. The segmentation results are further represented as closed polygons, facilitating further spatial analysis and shape relations. The inferred heatmaps directly encode the positional likelihood of the anatomical keypoint. The equivalent *x* and *y* coordinates are extracted by a spatial argmax operation or by a least-squares curve fitting in case predicted heatmaps suffer from large local intensity variation. The heatmap representation of an elongated structure is converted to a polyline by the following steps. First, the heatmap is interpreted as a 2D point cloud, and each point is assigned an importance weighting based on the predicted likelihood. By selecting the points with a weight above the threshold of 1σ, we extract only the most descriptive points for the target structure. Finally, we obtain the 2D polyline by interconnecting a set of equidistant points sampled from a smooth parametric B-spline which interpolates the selected points. If we are interested in a particular region of an object’s contour C, we alternatively interpret the structure’s heatmap as a weighted mask that reduces the contour to a certain subsection $C^{\prime }\subset C$.

#### 3.1.4. Stage C) Geometric Construction of Individual Planning Steps

Once the geometric descriptions and groupings are created, they are interrelated based on the planning geometry provided in the planning blueprint (Appendix B). According to the manual workflow, this interrelation is distributed over several steps, each targeting a single anatomical structure. The final result is then overlaid on the input image to provide visual guidance for the user. After close inspection and visual alignment of the planning to the underlying anatomy, the user might want to modify some parts of the planning geometry retrospectively. The design choice of our multi-stage approach naturally supports this use case. After manually adjusting any of the contained geometric objects, the geometric calculations in Stage 3) can be repeated at little computational cost, enabling real-time visualization of the planning on live images.

### 3.2. Multi-Task Network Architecture

The deployed multi-task network architecture (Figure 3 and Figure A3, Table A1) is a Fully Convolutional Network and builds on the highly symmetrical Hourglass architecture [54,76]. An Hourglass first captures and encodes the input morphology at various scales and abstraction levels in a contracting path. The encoded features are then processed in an expanding path and recombined with residual information from the encoder (via parameterized skip connections) to reconstruct a multi-channel representation of the input. This representation is the input to one ore more prediction heads, each responsible for a parameterized mapping to the desired output for one of the tasks.

Every convolution block in the Hourglass, both in the contracting path, the expanding path(s), as well as the task-specific mappings, is constructed as a stack of residual pre-activation bottleneck units [76]. These residual units introduce an additional identity function to provide a direct and unchanged propagation path for the features and allow the preservation of the feature identity across multiple processing steps [77,78,79]. Based on a joint representation of feature information at every stage in the computation graph, the network can learn an optimal iterative refinement strategy and globally retain important information without sacrificing information gain [79]. We argue that this design choice is particularly advantageous in MTL scenarios, where some tasks may put more emphasis on global semantics provided by highly aggregated abstract features (e.g., semantic segmentation). In contrast, others may depend more on low-level feature representations (e.g., precise anatomical keypoint localization).

In comparison to the U-Net topology, where the number of convolutional kernels typically increases with decreasing feature size [80], the Hourglass assigns equal importance to all abstraction levels by using a constant number of kernels throughout the complete encoder–decoder architecture.

### 3.3. Dataset, Ground Truth, and Augmentation Protocol

#### 3.3.1. Cohort 1: Diagnostic X-ray Images

First, we use diagnostic radiographs to analyze the effect of the MTL network topology, different loss/task weightings, and the amount of training data. We perform the ablation study on 211 (strictly) lateral diagnostic X-ray images of the knee joint acquired prior to reconstruction surgery. The images are of various spatial resolutions and occasionally show overlapping surgical implants. For each image, the study’s first author annotated the outline polygon for the femur (S1) and tibia (S2) bone using the labelme annotation tool [81]. The anatomical keypoints (K1–K7) and lines (L1–L3) that are part of the planning geometries for MPFL, PCL, and ACL reconstruction surgery were annotated by the study’s first author (training) an orthopedic surgeon (test) with a proprietary interactive tool by Siemens Healthcare GmbH. The data were separated into three subsets for training (167), validation (16), and testing (38) using stratified sampling. For the test split, all data that show a fixed-sized steel sphere with a 30 mm diameter was selected, allowing conversion between pixel and mm spaces.

#### 3.3.2. Cohort 2: Intra-Operative X-ray Images

We study the applicability to intra-operative surgical planning using 89 lateral intra-operative fluoroscopic X-ray images from 43 distinct patients. The retrospectively collected data were originally acquired during reconstructive trauma surgery with a mobile C-arm system with flat panel detector (CIOS Spin, Siemens Healthcare GmbH). Compared to diagnostic radiographs, this cohort poses unique challenges to the automatic image analysis system. The images show a considerable overlap of the anatomy with implants and surgical tools, variance in administered dose, and in some cases, bilateral exposure of the knee joints due to spatial limitations in patient positioning. Additionally, the field of view of mobile C-arms is smaller than that of conventional X-ray machines, which leads to a more pronounced truncation of the bone shafts. The cohort was split into training (60), validation (14), and test (15) with no patient overlap across subsets. Similar to the diagnostic data, the planning geometry for MPFL, PCL, and ACL reconstruction surgery was annotated by the first author using a custom extension of labelme.

#### 3.3.3. Augmentation and Ground Truth

For the training subset we applied online-augmentation with light spatial shearing (s∈[−2,2], p=0.5), horizontal flipping (p=0.5), in-plane rotation (α∈[−45,45], p=1), and scaling (s∈[0.8,1.2], p=1) [82]. Any spatial transformation of the ground truth data were performed using projections of the numerical coordinate representations to prevent information loss by discrete interpolation. Before network propagation, the image intensities were normalized to the value range of [0,1] using min–max normalization given by $\tilde{x}=\left(x-\min\left(x\right)\right)/\left(\max\left(x\right)-\min\left(x\right)\right)$. Since the original images are of different spatial resolutions and aspect ratios, they were brought to common dimensions of [H:256×W:256]px by bi-cubic down-sampling. Thereby, the original aspect ratio was maintained, and the images were zero-padded to the target resolution.

After augmentation, the numerical ground truth was converted to its respective spatial representation. For that purpose, the bone outline polygons were converted to binary segmentation masks with the original image resolution. The numerical keypoint coordinates were translated to spatial activation maps/heatmaps by sampling a bivariate Gaussian function without normalization factor (i.e., $(x_{0},y_{0})=1$ at keypoint coordinate) and a standard deviation of σ=6px. Similarly, the line heatmaps were generated by evaluating the point-to-line distance of the line-surrounding points in a locally-bounded hull. The line-orthogonal width of this hull was defined as w=6σ. The point and line heatmaps were truncated at ±3σ, the remaining values were set to zero.

### 3.4. Training Policy and Implementation Details

The network parameters of the pre-operative models were optimized for 400 epochs using mini-batch gradient descent with a batch size of 2 and Nesterov momentum [83]. The learning rate and momentum were adjusted according to the cyclical learning rate policy (CLR) [84], annealing both values linearly between a lower and upper bound (${\mathrm{lr}}_{\min}=0.001$, ${\mathrm{lr}}_{\max}=0.1$, ${\mathrm{momentum}}_{\min}=0.8$, ${\mathrm{momentum}}_{\max}=0.9$) within a full cycle length of 360 update iterations. For the intra-operative data, the training time was increased to 500. To overcome initially observed optimization instability, the cyclic gradient descent policy was replaced with RMSProp and a single learning rate decay with factor 0.1 at 350 epochs. A linear combination of Soft-Dice and Binary Cross Entropy $L_{\mathrm{seg}}\;:=\;\mathrm{sDice}+\mathrm{BCE}$ was used as loss function for the segmentation task. For keypoint and line regression, mean squared error losses $L_{\left\{\mathrm{kpt},\mathrm{line}\right\}}\;:=\;\mathrm{MSE}$ were used. In the case of dynamic updates of the task weights using Gradient Normalization, the task weights were optimized using the same update policy and the asymmetry hyper-parameter was set to α=1, which was found to be the optimal value in a previous study ([54]). The model selection was based on the minimum inference error across all tasks on the validation set, which we evaluated after each training epoch. The selection criterion was defined as the minimum unweighted composite loss, i.e., $L=\sum_{t=1}^{T}L^{t}$. The network parameters were penalized with an L2 weight regularization term of d=0.00005 to mitigate overfitting effects on the training subset.

All models were implemented in PyTorch v1.8 (Python v3.8.12, CUDA v11.0, cuDNN v7.6.5). It was ensured that all pseudo-random processes were initialized using the same random seed.

### 3.5. Evaluation Protocol

The sensitivity and precision of the predicted segmentation masks was measured using the Dice coefficient across both spatial dimensions, averaged over foreground and background. To also account for contour fidelity, we calculated the Average Symmetric Surface Distance (ASSD) [85]. For two contours *C* and $C^{\prime }$ with points x∈C and $x^{\prime }\in C^{\prime }$, the ASSD is computed by evaluating the distance function $d\left(x,C^{\prime }\right)=\min_{x^{\prime }\in C^{\prime }}{∥x-x^{\prime }∥}_{2}$.
$$\mathrm{ASSD}=\frac{1}{\left|C\right|+\left|C^{\prime }\right|}\left(\sum_{x\in C}d\left(x,C^{\prime }\right)+\sum_{x^{\prime }\in C^{\prime }}d\left(x^{\prime },C\right)\right)$$

The keypoint localization error is measured as the Euclidean Distance (ED) between the ground truth coordinates and network predictions. For the lines, we are interested in the quality of fit between centroids and the orientation angles. For that purpose, we first truncate the heatmaps at the likelihood of 2σ to get rid of outliers and then calculate their centroid $c_{x,y}$ and orientation angle γ via raw image moments $M_{pq}$ [86].
$$c_{x,y}=\left(\bar{x},\bar{y}\right)=\left(\frac{M_{10}}{M_{00}},\frac{M_{01}}{M_{00}}\right)$$
$$\begin{array}{cc} \gamma =\frac{1}{2}\arctan\left(\frac{2\mu_{11}^{\prime }}{\mu_{20}^{\prime }-\mu_{02}^{\prime }}\right),\;\mathrm{with}\;\;\mu_{20}^{\prime } & =\frac{M_{20}}{M_{00}}-{\bar{x}}^{2}\\ \mu_{02}^{\prime } & =\frac{M_{02}}{M_{00}}-{\bar{y}}^{2}\\ \mu_{11}^{\prime } & =\frac{M_{11}}{M_{00}}-\bar{x}\bar{y} \end{array}$$

## 4. Results

### 4.1. Research Questions

As motivated in Section 3.1, we analyze the proposed MTL algorithm for optimization of several anatomical detection tasks for 2D surgical planning. Specifically, we answer the following research questions:Rq (1).How does the MTL network topology and task weighting strategy affect anatomical feature extraction?Rq (2).Does sharing tasks across anatomically related applications improve the feature extraction and target positioning compared to the single-application variant?Rq (3).How does the number of training data affect the planning accuracy?Rq (4).Can the performance on highly-standardized diagnostics images be applied to more complex imaging data in the intra-operative environment?

### 4.2. Rq (1) Network Topology and Task Weighting

We study the effect of a shared inductive bias via different levels of hard parameter-sharing on the optimization behavior and surgical planning quality. The two different MTL topologies, multi-head and multi-decoder, are optimized using either a uniform or dynamic GradNorm task weighting. The performances are compared to the respective STL variant.

The individual prediction errors per application and anatomical feature according to *Stage A* of the multi-stage algorithm are illustrated in Figure 4. For all three application types (Figure 4a,c), the segmentation quality does not benefit from a shared feature representation. The difference between the STL variant and the unweighted MTL variants is negligible but increases substantially for the dynamically weighted variants, which is reflected most notably in the average contour error. Interestingly, this trend is more pronounced for the applications with three tasks (MPFL, PCL) than it is for the ACL application with two tasks.

In contrast, the median precision of the predicted keypoints is significantly higher for the task-weighted MTL variants and increases with the number of landmarks. Starting with the PCL application ($n_{\mathrm{kpt}}=1$) where no substantial difference in the localization error between the variants can be observed, the weighted multi-head and multi-decoder models lead to a reduction of the position error by up to 50% for the ACL application ($n_{\mathrm{kpt}}=5$). In the latter ACL case, this particularly shows for the tibial keypoints K4–K6 that are anatomically more complex and have a larger positional variance. There, the STL variant and unweighted multi-decoder variant suffer significantly from a comparatively large task-specific inductive bias. The reduced keypoint precision for these variants is accompanied by a drastically reduced maximum intensity of the predicted heatmaps, indicating low intra-model confidence and leading to error-prone intensity-peak extraction (Table 2, Figure 5). For the estimation of the lines, the results do not show a clear trend, but the weighted multi-head variant generally achieves the most accurate and consistent results. The results further indicate that in most cases, separating the reconstruction with task-specific decoders leads to slight improvements of the segmentation, but in all other cases is comparable to or even performs worse than the significantly more sparse multi-head variant. This observation suggests that not only a shared encoding but also a shared decoding of aggregated features with only a small trailing task-specific part is preferable.

These insights are in large part supported by the applications’ planning metrics calculated at *Stage C* of the multi-stage algorithm (Figure 6). Looking at the tibial aspects of the ACL planning, we see highly significant improvements of the approximated bundle attachments using a weighted MTL scheme compared to STL and unweighted MTL. However, the difference between the two levels of parameter-sharing is not significant, which means that a shared decoder neither benefits nor worsens the discriminative power. Interestingly, the planning metrics corresponding to the position of the femoral ligament attachment bundles are not positively affected by any MTL setting, with the single-task performance being among the best variants. This relation supports the previous findings (Figure 4) where anatomically intricate and ambiguous landmarks significantly benefit from more constrained and generalized features, and clearly visible ones, such as the Blumensaat keypoints K1 and K2, profit from a higher number of task-specific features, allowing to align much stronger to the most frequent anatomical pose.

In this context, it should be noted that the visible spatial layout of the proximal tibia head, where most considered landmarks are located, strongly varies with slight changes in the viewing direction (Figure A1). Since the complex 3D structure of the tibial articular surface cannot be fully assessed by a single projection to 2D, different viewing directions result in a greater positional variance of the landmarks. This variance leads to ambiguity in ground truth annotation and consequently limits the models’ ability to generalize well. The ambiguity is further increased by difficulties in interpreting depth information, caused by the general thickness of the tibial plateau, multiple intra-bone overlaps, and the resulting fuzzy contours. However, these uncertainties in landmark detection are less pronounced for the distal femur. There, the anatomy is less complex, and the depth in 2D projections is easier to assess. Moreover, the position of the femoral keypoints K1 and K2 located in the intercondylar notch is not affected by some rotation of the viewing direction around the Blumensaat line.

For the MPFL and PCL planning applications, the different network topologies and weighting variants do not significantly differ from each other. For MPFL planning in particular, all model variants meet the standard clinical requirements of (<2.5 mm) precision [21] and previously published inter-rater performance [54].

### 4.3. Rq (2) Combining Tasks across Multiple Surgical Applications

We analyze whether task sharing across different surgical planning types improves over the single-application case. For that purpose, we aggregate the anatomical features of a task for all considered planning applications and define a combined model that we analyze the same way as in Experiment (1).

As can be seen in Figure 4 and Figure 5, the precision of the keypoints estimated with the combined model benefits from a generalized feature set. Especially for the keypoints K3–K7 that suffer from positional uncertainty in the 2D projections, the combined model suffers from extensive spatial smearing at the heatmap peak and severely reduced likelihood (Table 2). This imprecision of the intensity’s spatial peak negatively impacts the argmax-based coordinate extraction, resulting in large deviations from the ground truth. Furthermore, the combined model’s segmentation quality slightly improves when using task-specific decoders, matching the models’ behavior in the single-application case. However, this performance gain does not translate into better line and keypoint prediction accuracy, which ultimately does not warrant the additional computational cost of multiple decoders. The combined model’s planning metrics show no significant change in performance compared to their single-application counter-parts in ACL and PCL planning (Figure 6 and Figure 7). However, the localization of the MPFL Schoettle Point suffers from a combined task definition which leads to larger angulation and general positioning errors of the approximated femoral shaft line (Figure 4).

### 4.4. Rq (3) Effect of the Number of Training Data

We analyze how the amount of training data influences the choice of MTL topology design and loss weighting. For that purpose, the size of the training cohort is subsequently reduced by taking only a fraction of data f∈{1.0,0.75,0.5,0.25} (random selection, fixed between experiments).

Almost all single-application variants (Figure 8a,c) evidence a low negative correlation of the planning errors with increasing data count (strictly negative correlation for all weighted MTL models). Most noticeable is the significant positive impact (correlation probability p<0.05) of the amount of data when applying the STL models to the ACL application (Figure 8b). However, the performance peaks are not always reached using the maximum amount of data. Although the femoral planning metrics, for example, benefit from a broader selection of data samples, the error rate is lowest at a reduced data factor of f=0.75.

No clear trend of the effect of differing amounts of training data can be extrapolated for combined task representations of multiple planning applications (Figure 8). For the STL models, the performance drops significantly with increasing data points, particularly for the femoral planning metrics. Interestingly, the performance of the MTL models remains almost unchanged with increasing factor *f*, indicating that a higher variance in the training distribution can not be exploited for learning a more generalized feature set. However, the direct comparison between STL and MTL in the multi-application scenario attests to the MTL variant a more effective extraction of the relevant features and thus a more stable learning process and better inference performance.

### 4.5. Rq (4) Application to Intra-Operative Data

We apply the weighted MTL algorithms to a complex intra-operative setting. The X-ray images show difficult trauma cases with fractured anatomy, strong instrumentation overlap, weak contrast, and bilateral exposure of the knee joints. We skip the evaluation of the PCL planning since it requires explicit knowledge of the pixel-to-mm relation.

For MPFL planning, the automatic planning achieves high accuracy for cases with clearly visible bone shaft (Table 3, Figure 9). In the case of strong shaft truncation, the estimated femoral cortex line suffers from dorsal shifting with increased deviation from the anatomical bone axis. The translation leads to malpositioning of the Schoettle Point. For extreme cases, the cortex line cannot be recovered. This dependency confirms previous findings highlighting the importance of an application-specific acquisition protocol, especially for the limited field of view in mobile C-arm imaging [50]. Although femoral ACL planning is highly accurate, tibial ACL planning suffers from ventral migration of the tibial proximal epiphysis corner towards the champagne drop-off point. This area is often superimposed by metallic plates, obscuring salient visual cues needed for precise localization. In 3 out of 15 test cases, the planning was cancelled due to fragmentation of the tibia segmentation mask and subsequent errors in contour extraction. These error modes can be reduced for three out of four bundle insertion points in ACL planning when using a multi-decoder model.

The increased complexity in the intra-operative setting could to be compensated by additional training data. In particular, more variation in instrument type and position must be covered to address the ambiguity in bone segmentation and the resulting high contour errors.

## 5. Discussion

Careful preparation of the surgical plan and its integration into the intra-operative workflow are key drivers for safety and low revision rates in reconstructive orthopedic surgery. Currently, available software mainly supports pre-operative planning on diagnostic 2D and 3D data and the following registration on intra-operative images. However, for some surgical interventions, such as ligament reconstruction and intramedullary nailing, pre-operative 3D scans are typically not acquired. Moreover, the usefulness of planning on static 2D radiographs remains questionable due to an unpredictable patient and device positioning during intra-operative management. A planning platform operating on intra-operative data would elevate surgical safety, increase the precision in critical positioning steps, and enable live outcome assessment in the operating roomt (OR).

In this study, we developed and analyzed a multi-stage algorithm that combines deep learning-based anatomical structure localization on 2D X-ray images and geometric post-processing to enable pre- and intra-operative surgery planning. The algorithm mimics the clinically established, manual planning rationale by separating feature detection and their geometric interrelation in a surgical plan. In the method evaluation on three trauma-surgical applications, we observed a high spatial precision in localizing optimal drill insertion points and low angulation errors for k-wire instrumentation and anatomical axis detection.

A fundamental characteristic of the multi-stage design is that the extrinsic positioning errors depend solely on the performance of automatic feature detection. While we cannot reduce the intrinsic error between the true anatomical position and approximation of the target by the planning blueprint, we approach extrinsic error reduction in two ways. First, users can manually adjust the position of the detected anatomical features, granting granular real-time control over the plan. Second, the feature extractor’s learning policy can be improved using an MTL scheme. MTL’s main rationale is to exploit similarities in data and underlying data formation by using a meta-learner that optimizes a common feature set. Our results substantiate the benefit of these generalized features for detecting anatomical structures on orthopedic image data. In particular, the precision of anatomically complex keypoints benefits from MTL compared to STL. We conclude that additional supervision of a keypoint’s context area reduces positional uncertainty by exploiting strong spatial correlations between the keypoint and the bone outline on a 2D projection. This outline can be reliably detected and exhibits low variation between patients. Since most radiographic keypoints lie on this outline, boundary-aware tasks, such as bone segmentation, provide strong auxiliary cues during optimization.

Consistent with recent MTL literature, task balancing effectively prevents overfitting to the segmentation task, which is spatially more dominant than the keypoint heatmaps and contributes high-magnitude gradient signals to the composite loss function. Although balancing with GradNorm shows superior performance over unweighted optimization, it solely targets homogenization of the tasks’ relative learning rates, not taking into account complex and potentially conflicting task interaction. We believe that it could be advantageous to combine this balancing with an active alignment of gradient direction and magnitude, e.g., by using Gradient Sign Dropout [74] or Gradient Vaccine [73]. Although such iterative improvements are attractive, their actual expediency is challenged by an upper performance limit imposed by the number of tasks, their concrete characteristics and relatedness, and the model complexity of the meta-learner.

This limitation is particularly evident when combining tasks across similar surgical applications. Faced with increased task complexity, the combined model is unable to find a common representation that improves over the single-application case. Without increasing model capacity—which comes at the expense of increased execution time and computational footprint—we advocate an MTL scheme on the application level rather than grouping multiple applications per anatomical region.

No clear conclusion can be made for the necessary amount of training data. Although the model performance increases slightly with more data samples, the magnitude of improvement neither warrants the additional labeling effort nor lets us define lower and upper bounds for the cohort size. In addition, the ablation study was performed on pre-operative diagnostic X-ray images with a highly standardized protocol for acquisition and patient positioning. This little variation compares poorly to the types of variance that are observed in the inter-operative environment. There, intricate fracture patterns, a broad array of surgical tools and implants, and greater out-of-plane rotation angles caused by spatially-constrained patient positioning introduce novel types of image characteristics. In this scenario, additional data may help to reduce model bias.

We see several limitations of our study. At the present stage, our method does not consider the registration of an initial surgical plan on subsequent intra-operative live data. Although the lack of registration does not immediately impact the planning process, it hamstrings outcome evaluation and more advanced real-time guidance using fluoroscopy or periodical positioning checks with single shots. Furthermore, the method was only evaluated for surgical applications on the knee joint. Compared to the large femur and tibia, the bones of the wrist, elbow or foot are much smaller and have less pronounced contours in the 2D projection due to strong inter-bone overlap. A further restriction of our method is the disconnect between the optimization criterion and the actual planning target. The algorithm’s first stage solves the anatomical feature detection problem independently from the planning geometry, which can be seen as optimizing a proxy criterion with less anatomic and geometric constraints. However, such constraints could help rule out implausible configurations and likely push the model to learn an abstract semantic representation of the actual planning target.

To address these shortcomings, future studies could investigate a combination of anatomical feature extraction and subsequent plan generation with direct regression of the planning target. The planning geometry can be implemented with differentiable functions, allowing gradient propagation from the actual target to the anatomical feature extractor. In this scenario, a consistency term between the geometric and direct target estimates can help align both estimates during optimization. Furthermore, previous studies have shown that planning methods are highly susceptible to rotational deviations between the underlying anatomy and the standard projection [87,88]. We believe that extending our method with an automatic assessment [89,90] of this mismatch could assist in preventing target malpositioning and improve the planning’s reliability and clinical acceptance. Another desirable extension is to combine our method with surgical tool recognition. In particular, knowledge of the k-wire direction and tip position would allow software-side security controls and advanced guidance in the intra-operative execution of the plan [56].

However, already in its current stage, the presented approach offers a precise automatic tool to support surgical intervention planning directly on intra-operative 2D data. Such intra-operative plan generation is particularly relevant for trauma surgery, where pre-operative 3D images are often unavailable, limiting the application of many analysis and navigation solutions operating on MRI [91,92,93,94] and CT [95,96,97] data. The proposed multi-stage planning algorithm was designed to separate automatic image analysis and geometric plan generation, allowing the user fine-grained control over each step of the proposed plan. This design decision ultimately improves user acceptance of the planning tool by allowing inaccuracies in the automatically proposed plan to be corrected for complex and rare injury patterns. Furthermore, we were able to show empirically that parallel detection of all anatomical features relevant to the plan using a weighted MTL policy benefits positioning accuracy, confirming the long-standing consensus in the MTL literature.

## 6. Conclusions

In this work, we specifically address the need for accurate X-ray guided 2D planning in the intra-operative environment for common injury patterns in reconstructive orthopedic surgery. The proposed multi-stage algorithm separates automatic feature extraction and the actual planning step, which we consider a pivotal design choice that allows for real-time control of individual planning components by the user. The detection step can be efficiently solved with a lightweight MTL scheme, allowing short execution times and improving spatial precision over the STL case. We believe that the developed platform will help overcome the limitations of current clinical practice, where planning is done manually on diagnostic 2D data with limited intra-operative value. Generating the surgical plan directly in the OR will elevate surgical accuracy and ultimately motivate the development of novel 2D planning guidelines. Future work should explore the possibilities of end-to-end training of the feature extractor to provide gradient updates in each step of the planning geometry. Additionally, integrating the method with automatic mechanisms for image alignment and surgical tool localization would assist the surgeon not only in planning, but also in image data acquisition and directly in surgical execution.

## Figures and Tables

**Figure 1 jimaging-08-00108-f001:**
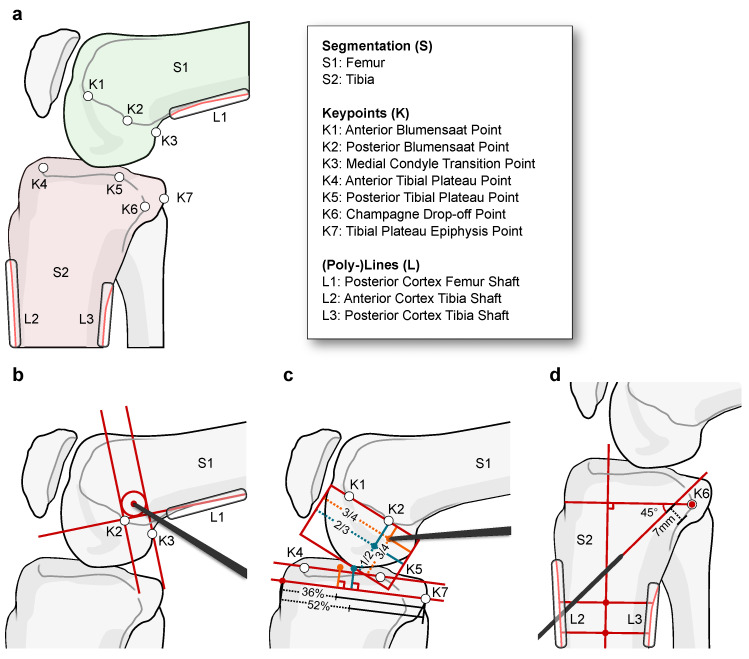
Illustration of the three considered surgical planning guidelines for ligament reconstruction on the knee joint. (**a**) Description and localization of relevant anatomical structures for all plannings. (**b**) Localization of the anatomic femoral insertion site for Medial Patellofemoral Ligament (MPFL) reconstruction surgery. The construction builds upon the clinical study by Schoettle et al. [21]. (**c**) Identification of the surface attachment points for the anteromedial (AM; orange) and posterolateral (PL; teal) bundles for Anterior Cruciate Ligament (ACL) reconstruction surgery. The planning follows the quadrant method by Bernard et al. [32] and Stäubli/Rauschning [34] for the femur and tibia, respectively. (**d**) Detection of the optimal drill path angulation and position for transtibial reconstruction of the Posterior Cruciate Ligament (PCL) according to Johannsen et al. [23].

**Figure 2 jimaging-08-00108-f002:**
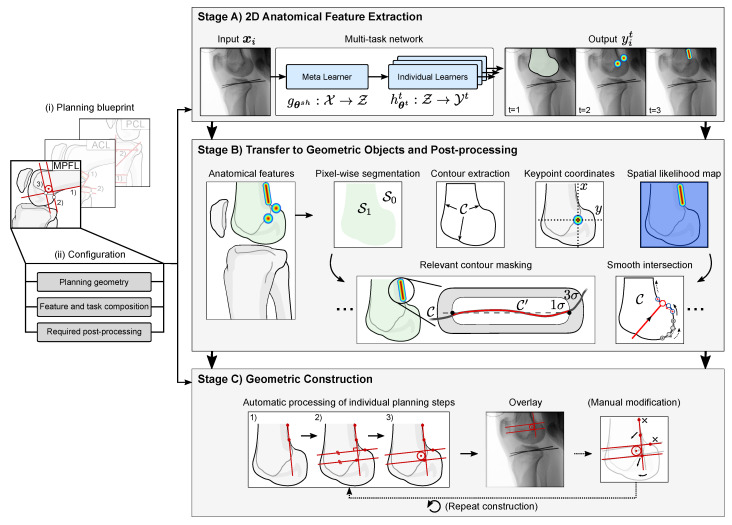
Overview of the automatic planning workflow. After selecting a planning blueprint and a corresponding configuration, the target X-ray image is processed in three consecutive stages. After extraction the 2D anatomical features with a multi-task network, they are transformed to their corresponding geometric descriptors. The individual construction steps defined in the planning blueprint are automatically calculated by interrelating the extracted geometry (here, steps 1–3 in Stage C)). If needed, the user can manually modify the planning by adjusting individual control points upon real-time updates of the complete plan.

**Figure 3 jimaging-08-00108-f003:**
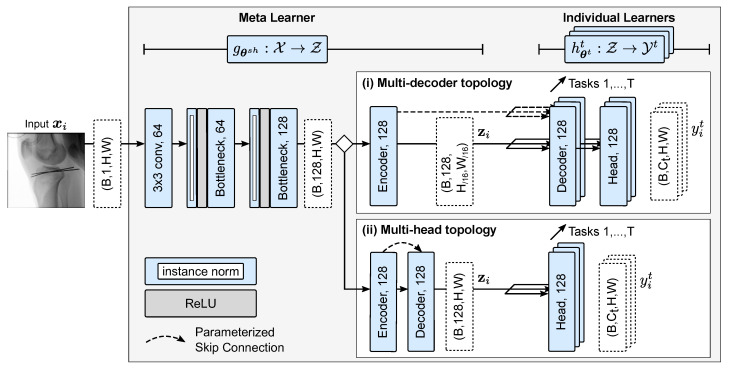
Architecture of the proposed multi-task network. After calculating a shared representation of the input image $x_{i}$ with the meta learner, an individual learner for each task computes an corresponding prediction $y_{i}^{T}$ in the task solution space $Y^{t}$. Two strategies for the individual learners with different levels of parameter sharing are displayed: (i) Multi-decoder topology with independent decoders and prediction heads, (ii) Multi-head topology with shared decoder but independent prediction heads. Details of the architecture are provided in Figure A3, Table A1.

**Figure 4 jimaging-08-00108-f004:**
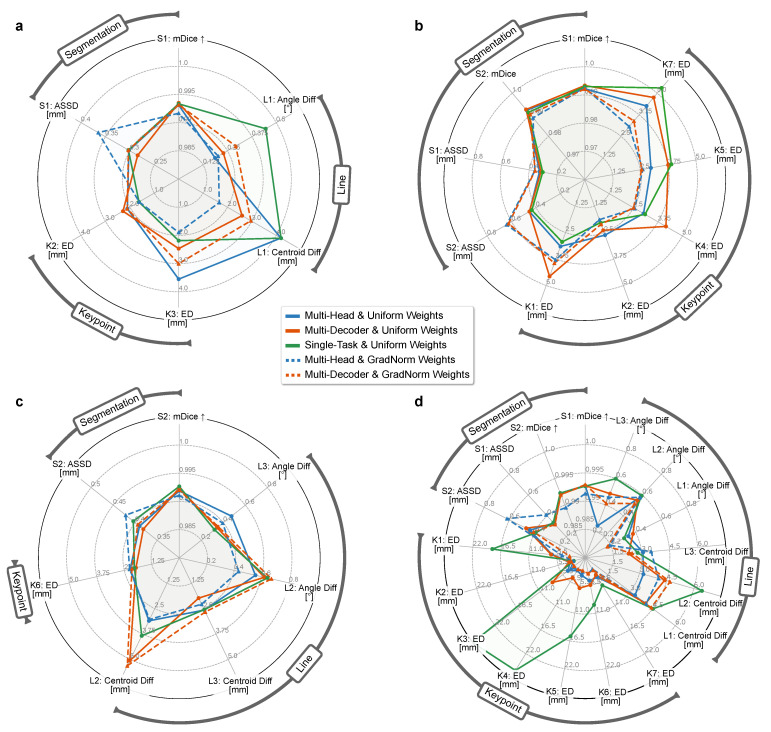
Radar charts of prediction errors per application and anatomical feature according to Stage A) (Section 3.1.2) and different model topologies and optimization policies. For all metrics but the mean Dice coefficient (mDice ↑) for segmentation tasks, lower is better. (**a**), MPFL. (**b**), ACL. (**c**), PCL. (**d**), combined (MPFL, ACL, PCL).

**Figure 5 jimaging-08-00108-f005:**
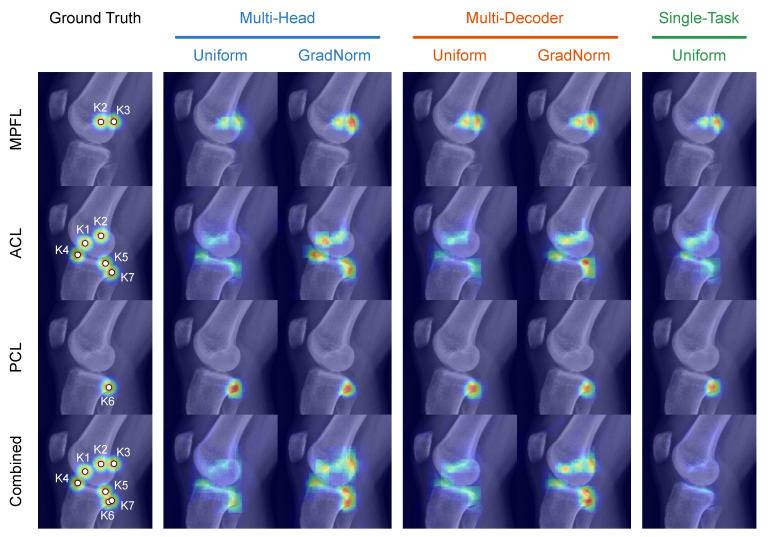
Visualization of heatmap prediction quality for different model topologies and task weighting strategies for a random sample from the test set.

**Figure 6 jimaging-08-00108-f006:**
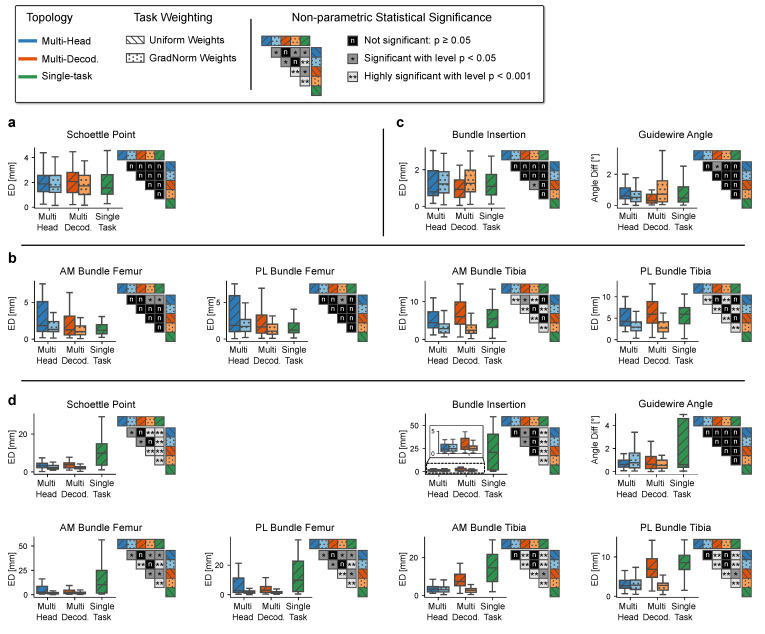
Boxplots of the planning metrics for each surgical application and different model topologies and optimization policies (without fliers/outliers). (**a**), MPFL. (**b**), ACL. (**c**), PCL. (**d**), combined (MPFL, ACL, and PCL). For each boxplot and topology pairing, we tested statistical significance with a two-sided Mann–Whitney U rank test ((n) not significant: p≥0.05, (*) significant: p<0.05, (**) highly significant: p<0.001).

**Figure 7 jimaging-08-00108-f007:**
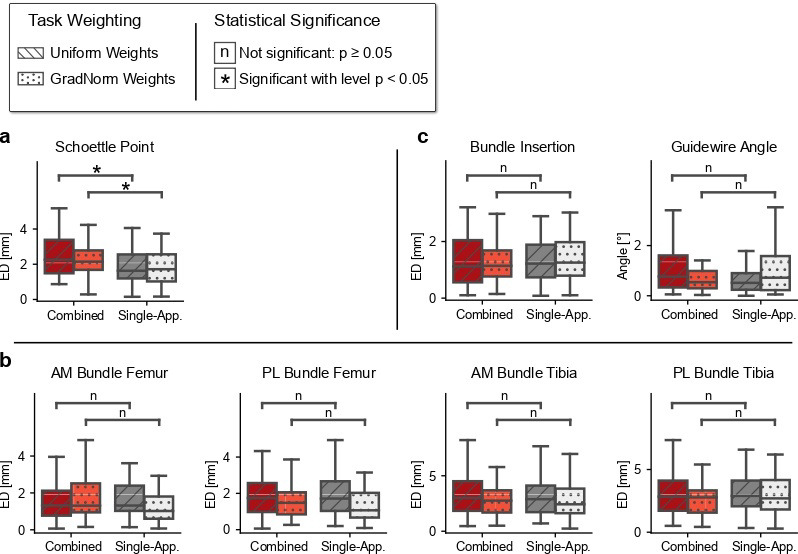
Comparison between single- and multi-application model performance on the test set. (**a**), MPFL. (**b**), ACL. (**c**), PCL. For each boxplot and pairing, we tested statistical significance with a two-sided Mann–Whitney U rank test ((n) not significant: p≥0.05, (*) significant: p<0.05.

**Figure 8 jimaging-08-00108-f008:**
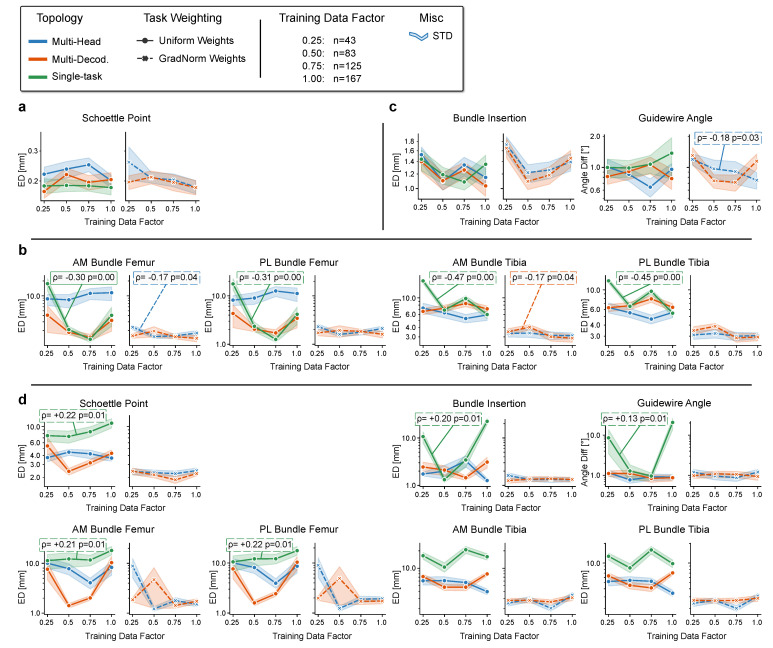
Effect of different amounts of training data on the planning metrics (logarithmic scale for y-axis for better visibility). (**a**), MPFL. (**b**), ACL. (**c**), PCL. (**d**), combined (MPFL, ACL, PCL). We report correlation coefficients ρ with rejection probability p<0.05 based on two-sided Spearman’s rank correlation measure.

**Figure 9 jimaging-08-00108-f009:**
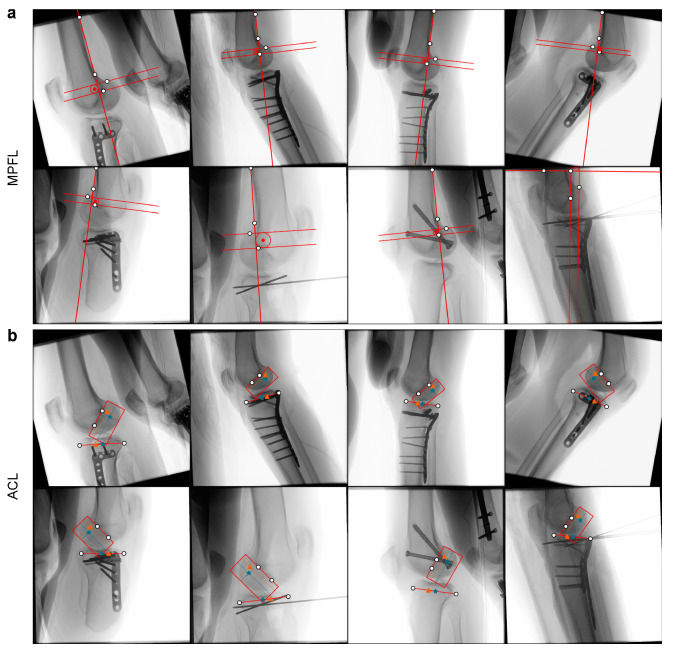
Planning predictions on intra-operative test data with multi-head topology and GradNorm task weighting. (**a**), MPFL. (**b**), ACL.

**Table 1 jimaging-08-00108-t001:** Mathematical representation of the relevant anatomical structures that allow subsequent optimization and geometric analysis.

Anatomical Structure	Spatial Representation
Semantically coherent regions	Pixel-wise and multi-label segmentation. The multi-label aspect allows for overlap-aware segmentation, e.g., in areas between bones or metal implants.
Anatomical keypoints	Individual heatmaps/activation maps. For that purpose, a multivariate Gaussian distribution with its mean at the keypoint coordinate and a predefined standard deviation is sampled.
Elongated structures	Line-symmetric heatmap/activation map. The distance to the line segment or the axis of interest is evaluated and transformed using a Gaussian function.

**Table 2 jimaging-08-00108-t002:** Analysis of the maximum intensity of keypoint heatmaps for different model topologies and task weighting strategies. Each variant’s values for mean and standard deviation (mean±std) are calculated across all application-specific keypoints on every test set image. The results of the best network variant for each planning methodology is marked in bold.

	Multi-Head		Multi-Decoder		Single-Task
Planning	Uniform	GradNorm	Uniform	GradNorm	Uniform
MPFL (*n* = 2)	0.64±0.11	0.83±0.09	0.76±0.09	0.76±0.11	0.80±0.12
ACL (*n* = 5)	0.42±0.08	0.71±0.12	0.43±0.07	0.73±0.12	0.57±0.16
PCL (*n* = 1)	0.90±0.11	0.84±0.11	0.83±0.09	0.79±0.10	0.84±0.08
Comb. (*n* = 7)	0.41±0.12	0.66±0.12	0.38±0.10	0.71±0.12	0.23±0.11

**Table 3 jimaging-08-00108-t003:** Numerical planning results on intra-operative test data. Each metric value is reported w.r.t. the original resolution of [H:976×W:976]px. The results of the best network variant for each planning methodology is marked in bold. In three ACL cases, planning was not possible due to poor segmentation quality.

		Multi-Head		Multi-Decoder	
Planning	Metric [px]	Median, ${\mathrm{CI}}_{95\%}$	Cnt.	Median, ${\mathrm{CI}}_{95\%}$	Cnt.
MPFL	Schoettle Pt.	9.08,[6.35,128.21]	15	14.02,[9.72,22.46]	15
ACL	AM Femur	8.17,[5.34,15.65]	12	4.80,[3.79,14.04]	12
	PL Femur	5.66,[4.17,12.84]	12	5.04,[2.85,12.48]	12
	AM Tibia	22.13,[9.52,82.04]	12	14.18,[10.27,42.87]	12
	PL Tibia	13.17,[10.09,38.74]	12	16.55,[12.85,41.71]	12

## Data Availability

The image data used for the evaluation of the software cannot be made available. The methods and information presented here are based on research and are not commercially available.

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
