# Peer review of "Multi-Stage Platform for (Semi-)Automatic Planning in Reconstructive Orthopedic Surgery"

_2313-433X, 2022, doi:10.3390/jimaging8040108_

Round 1

Reviewer 1 Report

This work shows how difficult semiautomated is as the author’s show that the proximal tibia could not be segmented reliably, whereas the distal femur could.  The authors also were noting that positioning was often difficult and time consuming.  These honest observations are extremely helpful.  Oftentimes papers attempt to prove they have solved all problems.  The authors do the subject a significant favor by “showing where the holes are” and where to direct future research attention.

Reviewer 2 Report

Dear Authors,

After reviewing your article I found it interesting and I think it can be interesting to Journal's readers. However, some minor revisions are recommended. Please see the attached document for details.

Kind regards,

Reviewer

Reviewer 3 Report

Very interesting work:

I have just one remark:

Please depict in details the best architecture of your deep learning system, i.e. number of layers, nodes etc.
